# An Interactive Teaching Tool Describing Resistance Evolution and Basic Economics of Insecticide-Based Pest Management

**DOI:** 10.3390/insects13020169

**Published:** 2022-02-04

**Authors:** Christian Nansen

**Affiliations:** Department of Entomology and Nematology, University of California, Davis, CA 95616, USA; chrnansen@ucdavis.edu

**Keywords:** resistance evolution, integrated pest management, insect population modeling

## Abstract

**Simple Summary:**

To attract bright, creative, and curious students to the academic fields of applied entomology and sustainable food production, instructors of undergraduate and graduate student courses should discuss experiences with lectures and lab sessions and share effective interactive teaching tools. This communication describes how a simple population model in an Excel spreadsheet can be used in teaching both insecticide resistance evolution and basic economics of insecticide-based pest management. A tutorial video and the model as an Excel spreadsheet are freely available. Through hands-on experience with this and similar interactive teaching tools, students will acquire fundamental knowledge about basic structures population models and they will acquire experience with quantitative data interpretation. Teachers can use this tool and accompanying tutorials to demonstrate how models can be used to describe and visualize complex interactions between insect genetics and crop management. Furthermore, data from published studies can be analyzed and discussed using this interactive teaching tool.

**Abstract:**

Effective teaching of complex concepts relies heavily on the ability to establish relevance of topics and to engage students in a constructive dialogue. To connect students with abstract concepts and basic theory, instructors foster and facilitate an engaging teaching environment. Population modeling is a cornerstone in applied entomology. However, it is also a topic and skill set that requires both basic mathematical and biological knowledge, and it may be perceived by students as being abstract and exceedingly theoretical. As a way to introduce entomology students at both that undergraduate and graduate levels to hands-on experience with population modeling, a well-established and widely used deterministic genetic population model is presented as an interactive teaching tool. Moreover, the general model describes three genotypes (SS = homozygous susceptible, SR = heterozygous, and RR = homozygous resistant) during 30 discrete and univoltine generations under a shared population density dependence (carrying capacity). Based on user inputs for each genotype (survival, fitness cost, reproductive rate, emigration, and immigration) and an initial resistance allele frequency, model outputs related to resistance evolution are produced. User inputs related to insecticide-based pest management (pest density action threshold, crop damage rate, insecticide treatment costs, and profit potential) can also be introduced to examine and interpret the basic economic effects of different insect pest management scenarios. The proposed model of resistance evolution and basic economics of pest management relies on a large number of important simplifications, so it may only have limited ability to predict the outcomes of real-world (commercial) scenarios. However, as a teaching tool and to introduce students to a well-known and widely used genetic population model structure, the interactive teaching tool is believed to have considerable utility and relevance.

## 1. Introduction

Most scientific journals publish standard research articles, short communications, opinion pieces, and reviews, but the number of articles describing aspects of scientific teaching and use of interactive teaching tools are rare or entirely absent. As research is either partially or directly linked to the capacity of students to become future scientists, educators, and decision makers, it is somewhat surprising that the teaching of science is not recognized as a widely accepted publication subject. Using the word, “teaching”, as a search term, I found a rather large body of short communication style articles published between 1910 and 1950, with just a few examples highlighted in [1,2,3,4,5,6]. The same literature search also suggested that, within the last 50 years, the word “teaching” in entomological science journals is almost exclusively restricted to obituaries of prominent entomologists with references to their teaching appointments. This communication is intended to promote a resurrection of a past publication priority, namely articles describing entomological teaching methods and resources. To attract bright, creative, and curious students to the academic fields of applied entomology and sustainable food production, it is paramount that we as instructors discuss experiences with lectures and lab sessions and that we share effective interactive teaching tools. As a way to introduce entomology students at both undergraduate and graduate levels to hands-on experience with modeling more broadly, a well-established and widely used deterministic genetic population model is presented as an interactive teaching tool. The basic teaching goal addressed with this interactive teaching tool is to provide students with an opportunity to examine consequences over time of basic coefficients influencing population dynamics. The proposed model is not intended as a tool to accurately predict the outcomes of real-world (commercial) scenarios. Moreover, it is intended to be a teaching tool to introduce students to a well-known and widely used model structure and to enable students to learn about both the advantages and limitations of genetic population models. Using this interactive teaching tool, two separate but highly complementary aspects of insect pest management can be discussed with students: (1) risk of insecticide resistance evolution and (2) basic economics of insecticide-based pest management. The examined coefficients may be obtained from published research articles or they may be selected based on certain assumptions, such as the allele being partially recessive or resistance being incomplete. I believe that it is particularly timely to resurrect this publication focus, as publications describing interactive teaching tools can nowadays be accompanied by links to tutorial Youtube videos and other learning resources.

## 2. Materials and Methods

The following describes an interactive teaching tool, in which a simple genetic population model in an Excel spreadsheet is used to model both insecticide resistance evolution and economics of insecticide-based pest management. Both students and instructors are strongly encouraged to contact me directly about the interactive teaching tool and to suggest improvements and changes to both the genetic population model itself and to accompanying tutorials. A tutorial video and the Excel spreadsheet are freely available via links on the following website: https://chrnansen.wixsite.com/nansen2/teachingtool (accessed on 1 February 2022). Due to possible updates in the future (new video links), it is recommended to access available tutorials through this website. Deterministic genetic population models to describe monogenic (and with two alleles = bialelic locus) resistance evolution to insecticides have been published over the past five decades [7,8,9,10], and the basic modelling framework has remained important [11,12,13,14]. The general genetic population model for each of three genotypes (SS = homozygous susceptible, SR = heterozygous, and RR = homozygous resistant) is described in Equation (1), and separate models for three genotypes are run for 30 discrete and univoltine generations under a shared population density-dependence (determined by a carrying capacity, *K*):(1)$$N_{x}\prime =W_{x}\times (N_{x}-E_{x}+I_{x})\cdot e^{\frac{Ln[R_{X}\times \left(1-F_{x})\;\right]\times \left(K-(N_{x}-E_{x}+I_{x})\right)}{K}}$$
where $N_{x}$ is the population size of genotype *x* in the current generation and $N_{x}\prime$ is the population size of genotype *x* in the following generation. Immigration (*I*) is set to occur prior to the reduction in survival (*W*) imposed by insecticide treatment (assumed to occur early in the generation time), while emigration (*E*) is set to occur after a reduction in survival (*W*) imposed by insecticide treatment (so assumed to occur late in the generation time). The parameters in Equation (1) are briefly described below and presented in light-green cells in Figure 1a:Carrying capacity (*K*): A theoretical pest population equilibrium density for a given “cropping universe” (i.e., crop field, greenhouse, orchard, etc.). A carrying capacity value is needed for the inclusion of density-dependent population dynamics. As default, *K* = 1000. However, it can be changed when applied to specific case stories of economics of pest management strategies, in which density of crop stand (and therefore insect pest carrying capacity) is assessed [15,16]. It may also be changed if pest population densities are expected to vary markedly among seasons as a function of crop management practices (i.e., crop rotation) and/or due to fluctuating seasonal weather conditions. Finally, the carrying capacity may be set to a specific value if it is known that a given crop field equals a certain number of plants per hectare and it is known that each crop plant can host a certain number of insect pest individuals.Resistance allele frequency: The initial frequency of allele(s) conferring insecticide resistance. With two alleles, initial frequencies of SS, SR, and RR genotypes are calculated based on assumption of the Hardy–Weinberg equilibrium. For simplicity, only two alleles and three genotypes are included. If three or more alleles or multiple genes are assumed to be involved and to maintain assumption of Hardy–Weinberg equilibrium, the six or more possible genotypes may be grouped into low, medium, high levels of (polygenic and/or metabolic) resistance levels [17]. In other words, this simple model structure can be used to examine and visualize expected outcomes from scenarios that are more complex than those involving only one gene and two alleles.Survival (*W*): Average survival of each genotype in response to insecticide treatment (0 = no survival and 1 = 100% survival). That is, insect pests have different survival rates for each developmental stage, but for simplicity, a single average survival coefficient for all life stages is used.Net reproductive rate (*R*): The natural logarithm of the offspring/adults ratio from one generation to the next, and a value of Ln(5) is proposed as default. Generally speaking, a high *R* coefficient shifts temporal trends left-wards (resistance evolution after few generations), while a low *R* coefficient shift temporal trends right-wards (resistance evolution rate).Fitness cost (*F*): A large body of literature describes how the evolution of insecticide resistance may be associated with a wide range of fitness costs [10,14,18,19,20,21,22,23,24]. For simplicity, only a single model parameter is included to encompass the total relative reduction in net reproductive rate (loss of fitness) due to insecticide resistance (0 = no fitness cost and 1 = complete loss of reproductive rate).Emigration (*E*): Number of pest individuals leaving the given cropping universe at the end of each generation. Emigration is calculated based on a fixed percentage of population density in the previous generation, *N*, of each genotype. Thus, if *E* = 10, then 10% of pest individuals of that genotype will emigrate. The fixed *E*-value for each generation does not take into account that pest individuals may leave at different time periods during the growing season and, therefore, cause different levels of crop damage. Moreover, the fixed *E*-value should be considered an average emigration rate for each generation.Immigration (*I*): It is assumed that insect pest individuals in the given cropping universe originate from a “global pest population” with a fixed genotypic composition. The genotypic composition of this global pest population is described by resistance allele frequency (cell C3 in the interactive teaching tool) and by assumption of the Hardy–Weinberg equilibrium. Thus, the genotypic composition of insect pest individuals during initial immigration as well as during immigration in each of the 30 generations is based on these fixed assumptions. The actual number of pest individuals immigrating into the given cropping universe in each generation is calculated based on a fixed percentage, *I*, of the carrying capacity, *K*. Thus, if *I* = 10 and *K* = 1000, then 100 insect pest individuals immigrate into the cropping universe each generation. Furthermore, these 100 immigrating insect pest individuals in each generation have a genotypic composition based on cell C3 in the interactive teaching tool. The effect of immigration is added to the model before a reduction in survival (*W*) imposed by insecticide treatment. The fixed *I*-value for each generation does not take into account that pest individuals may arrive at different time periods during the growing season and, therefore, cause different levels of crop damage. Moreover, the fixed *I*-value should be considered an average immigration rate for each generation. The fixed *I*-value may also be viewed as representing the effect of a non-insecticidal refuge in efforts to avoid/delay resistance evolution [25].

The deterministic genetic population model described in Equation (1) has been expanded further to also include parameters related to the modeling of basic economics of insecticide-based pest management (light-blue cells in Figure 1a). These parameters are derived from foundational articles about integrated pest management [26,27,28,29].

Economic injury level (EIL): As originally stated by Stern, Smith, Bosch and Hagen [29] and discussed in detail by Pedigo, Hutchins and Higley [26], EIL is defined as “the lowest population density that will cause economic damage”. In cell C10 in the interactive tool, an EIL value is: 0 ≤ EIL ≤ *K*. If EIL = 0, then mortality imposed by insecticide treatment (and selection pressure), *W*, is applied in all 30 discrete and univoltine generations. If EIL = *K*, then no mortality imposed by insecticide treatment will occur in ant of the 30 discrete and univoltine generations. An EIL of about 2-10% of *K* may be realistic starting point when developing and comparing model scenarios. For simplicity, it is assumed that pest population density and crop damage are directly (and linearly) correlated. As reviewed and discussed elsewhere in great detail [26], there are many important reasons for this correlation not to be considered linear and to vary among locations and growing seasons. However, as an introduction to basic economics of insecticide-based pest management, a linear and constant (fixed average for all life stages) response may in many cases be considered an acceptable assumption [26]. Accordingly, loss of survival, *W,* is only invoked in a given generation if total pest population density exceeds the user-defined EIL. In the interactive teaching tool, it is assumed that only a single insect pest is present, and it is exclusively responsible for all potential crop damage and therefore yield loss. If scenarios involving *Bt*-transgenic (*Bacillus thuringiensis*) crops are examined, then it would be assumed that mortality imposed by *Bt* toxins in crop plants is constitutive, so an EIL = 0 should be used.Damage rate: The relationship between insect pest density and crop damage (loss) is, for simplicity, assumed to be linear, and the slope must be >0. It is one of the most important quantitative aspects of insect pest management, and it was the main topic of a seminal article [26]. In cases where insect pests feed directly on what is harvested (i.e., fruits and seeds), direct crop damage is typically calculated as a direct function of the number of pest individuals. However, accurate calculation of the damage rate becomes considerably more complicated when insect pests cause indirect crop damage, which is also referred to as crop injury. Moreover, leaf herbivores or root-feeders cause stress of crop plants, so that plant injury indirectly leads to loss of yield. Additionally, the yield effect of such indirect crop damage varies as a function of crop stage and of which part of crop plants is injured, and there may be certain levels or intensities of injury leading to crop damage (damage boundary) [26]. Thus, many pest–crop scenarios encompass rather complex and frequently non-linear associations of insect pest density with injury, crop damage, and crop loss. Here, these associations have been markedly simplified, and a single and fixed parameter, damage rate (encompassing both crop injury and loss), describes the linear increase in crop damage as a function of insect pest density.Treatment cost (in US Dollars): total insecticide treatment costs of insecticide, labor, and use of equipment. It may also represent the relative cost of planting *Bt*-transgenic crops versus non-*Bt* crops. This parameter is used in well-described calculations of pest density action thresholds [27]. If the cropping universe is measured in hectares or acres, a large body of applied research articles and websites can provide specific information about the costs of insecticide treatment costs. Treatment cost does not take into account the secondary effects of insecticide treatments. Thus, possible adverse effects on natural enemies, outbreaks, or suppression of secondary pests are not factored into the simple economic calculations.Profit potential (USD): the potential value of crop harvest under the assumption of zero crop damage due to the specific pest and after excluding all non-insecticidal operational production costs. For simplicity, loss of profit as a function of insect pest-induced crop damage is assumed to be linear. Such a simplification does not apply to crop harvests, in which quality grading exists, and/or when a given harvest has to meet certain minimum specifications to be sold on niche—or restrictive—export markets. Finally, such a linear association between profit potential and crop damage does not take into account the possibility of damage by one insect pest potentially increasing the risks of secondary pest issues, such as higher risks of mycotoxins in moth-infested grain or nuts [30]. Thus, the basic economics portion of the interactive teaching tool does not include many of the important reasons why more advanced agricultural economic models are needed for reliable predictions of specific pest–crop systems. However, as general and basic introduction, it was found acceptable to use a highly simplified model structure and to assume linear association between crop damage and profit potential. In column M in the Excel spreadsheet, a logical statement is used to ensure that estimated profit potential cannot be negative.

## 3. Results

### 3.1. How to Use the Interactive Teaching Tool

Figure 1a shows the user interface, in which coefficients are introduced into cells in light green (resistance evolution) and light blue (basic economics of insecticide-based pest management) and with separate columns for each of the three genotypes. Based on input coefficients, three graphs illustrate the model outputs during 30 discrete and univoltine generations: (1) percentage of pest individuals for each of the three genotypes and total pest population as a percentage of the carrying capacity (Figure 1b), and (2) resistance allele frequency (Figure 1c).

The predicted profit potential is also visualized but will be described below. In the theoretical example presented in Figure 1, resistance was assumed to be partially dominant (also referred to as incomplete dominance) and is not associated with a fitness cost (*F* = 0 for all genotypes). Furthermore, it was assumed that 20% of homozygous susceptible individuals survive (W_SS_ = 0.20), and this additional survival is also added to partial dominance of heterozygous individuals (20% + 50% = 70%), so W_SS_ = 0.20, W_SR_ = 0.70, and W_RR_ = 1.00. Based purely on genetics, a given insecticide may be expected to kill 100% of homozygous susceptible individuals. However, some survival of these insect pest individuals could be attributed to factors accounting for low and incomplete spray coverage of contact insecticides [31,32] and/or sublethal and non-uniform concentration of systemic insecticides in portions of crop canopies [33]. In this theoretical scenario, the total pest population reaches 80% of carrying capacity after about 11 generations, and this coincides with the resistance allele frequency reaching 50%. Furthermore, the total pest population density exceeded 10% of the carrying capacity after five generations. The assumption of 20% survival by homozygous, susceptible individuals due to low and inconsistent spray coverage may be considered very high, and it may be considered possible to improve insecticide spray coverage and/or expression of insecticides in crop canopy. Thus, survival rates may be reduced so that survival by homozygous susceptible and heterozygous individuals is only 5% (W_SS_ = 0.05, W_SR_ = 0.55, and W_RR_ = 1.00), while other parameters are unchanged. Using these reduced survival rates, the total pest population density was 3.3% of the carrying capacity after five generations (figure not shown). Thus, this simple comparison of model outputs suggests that optimizing insecticide applications (minimizing the survival of homozygous, susceptible and heterozygous individuals) may reduce the total pest population density by several fold.

The scenarios described above show how the interactive tool can be used to examine basic hypotheses about survival rates due to insecticide spray performance. However, the scenarios also illustrate the fundamental dilemma regarding the development of sustainable pest management tactics based almost exclusively on the applications of insecticides [34]: that maximization of insect pest population suppression increases risk of resistance evolution. That is, the mortality of insect pest individuals in a given population is not random, and the application of insecticides imposes a selection pressure on the pest population. Only insect pest individuals that are genetically susceptible and exposed to a lethal dosage are suppressed. If 99.999% of an insect pest population is suppressed by an insecticide application, then only very few (0.001%) but highly resistant individuals will survive. Thus, the next insect pest generation will have much fewer pest individuals, but they will all be resistant. Additionally, under an assumption of complete resistance and constant insecticide-induced selection pressure, the population density of these resistant individuals will grow over time. The interactive teaching tool can be used to explore and identify possible scenarios, in which both total pest population density and resistance allele frequency remain low over time. This objective may be virtually impossible to achieve under commercial and real-world conditions and is therefore one of the fundamental drivers behind the development and adoption of integrated pest management. It has been my personal teaching experience that presenting this interactive tool to students and engaging them in the development and interpretation of different scenarios is one of the most effective ways to enlighten them about the importance of promoting integrated pest management as a sustainable approach to food production and pest management.

### 3.2. Interactive Modeling of Resistance Evolution

Several decades of genetic modeling of risks of resistance evolution to transgenic *Bt*-transgenic crops has led to a widely accepted “high-dosage refuge strategy” to avoid/delay resistance evolution in target pest populations [23,25,35,36]. The high-dosage component of this strategy implies that *Bt* toxin expression must be high enough to suppress virtually all homozygous and susceptible, and all heterozygous insect individuals within the target pest population in a given cropping universe. The refuge component of this strategy implies that homozygous, susceptible individuals develop in an adjacent non-*Bt* crop and mate with the few homozygous, resistant pest individuals surviving from the *Bt*-transgenic crop. Based on extensive genetic population modeling, it is widely accepted that high-dosage refuge strategies are only sustainable if [23,25] (1) refuges are sufficiently abundant (in the interactive teaching tool, this is directly linked to an immigration rate of SS into the cropping universe), (2) allele(s) conferring resistance is/are rare, (3) the resistance is recessive (therefore, W_SR_ much smaller than W_RR_), (4) fitness costs are associated with resistance, and (5) the resistance is incomplete (W_rr_ < 1). The interactive teaching tool can be used to experimentally examine all possible combinations of these important assumptions. In addition, it can be used to identify possible scenarios (combinations of coefficients), in which both total pest population density and resistance allele frequency remain low over time.

The following case story is based on published studies of corn earworm (*Helicoverpa armigera*), which is a serious polyphagous pest on many important crops. Furthermore, numerous studies have described corn earworm resistance to *Bt* toxins [36,37,38,39]. In the 2002/2003 growing season, the estimated Cry1Ac resistance allele frequencies in field populations of corn earworm in three geographical regions of India were 0.0007, 0.0011, and 0.0013 for the northern, central, and southern regions, respectively [40]. Based on a simple average, the Cry1Ac resistance allele frequency in 2002/2003 in India may be assumed to have been around 0.001. Kukanur, Singh, Kranthi and Andow [39] the examined resistance evolution in 2013 and 2014 from field populations of corn earworm in India and found the Cry1Ac resistance allele frequencies to be 0.085 in 2013 and 0.035 in 2014. Thus, during an 11-year period, there was a 58-fold increase in Cry1Ac resistance allele frequency to around 0.05. The interactive teaching tool can be used to adjust survival rate, fitness cost, and emigration and immigration rates of genotypes to produce model outputs depicting an increase in resistance allele frequency from 0.001 to 0.05 during an 11-year time period.

Figure 2a shows input the coefficients of a scenario, and Figure 2b shows the predicted resistance allele frequency (Freq_R) reaching 0.05 after 12 generations. This scenario was generated based on (1) an assumption of no/negligible refuges (so very low immigration rates and equivalent to 2% of the carrying capacity, because it was assumed that refuges were small and/or rare), (2) knowledge about the initial frequency of allele(s) conferring resistance (0.001), (3) the assumption that resistance is recessive (W_SS_ = 0.16, W_SR_ = 0.16, and W_RR_ = 0.80), (4) the existence of fitness costs associated with resistance and with the fitness cost being additive (F_SS_ = 0, F_SR_ = 0.13, and F_RR_ = 0.26), and (5) the assumption that resistance is incomplete (so W_RR_ < 1.00). Thus, what appears to be a reasonable scenario was found to align with field observations from India (the cell highlighted in yellow in Figure 2b). Furthermore, increasing the immigration rates of all three genotypes to 15% (which has the same effect as including a 15% refuge), the interactive tool predicted that a resistance allele frequency remains below 1% for 16 generations (Figure 3b). Moreover, the only difference between the scenario presented in Figure 2 and Figure 3 is the immigration rate (2% of the carrying capacity in Figure 2 and 15% in Figure 3). Thus, the interactive tool illustrates and corroborates the important recommendation of farmers growing a non-*Bt* refuge, as refuges lead to a marked delay in insect pest resistance evolution to *Bt*-transgenic crops. In many *Bt*-cropping systems, recommended non-Bt refuges make up 20%, which, compared with the model output shown in Figure 3, would further delay resistance evolution to *Bt*-transgenic crops.

### 3.3. Interactive Modeling of Basic Pest Management Economics

In the basic economics portion of the interactive teaching tool, resistance evolution values are slightly modified (Figure 4a). A scenario was established in which the intention was to approximate what could be envisioned as being realistic under commercial crop management conditions. Moreover, it is assumed that the spray coverage of insecticide applications is incomplete, so that 10% of homozygous susceptible pest individuals survive (W_SS_ = 0.10). It is also assumed that insecticide resistance is partially additive and incomplete, so W_SR_ = 0.45 and W_RR_ = 0.95. It is assumed that insecticide resistance is associated with an additive fitness cost (F_SS_ = 0, F_SR_ = 0.10, and F_RR_ = 0.20). The latter assumption has been documented in a wide range of insect pests and to a wide range of insecticides [19,20,22,24]. Agricultural statistics on processing tomato in California in 2019 show that the average production per ha was about 123 tons per ha, and the production value of one ha was about USD 9813 (https://www.cdfa.ca.gov/Statistics/PDFs/2020_Ag_Stats_Review.pdf) (last accessed 1 February 2022). A separate and very detailed study on the production costs of processing tomato in California in 2017 found total costs to be USD 8.283 per ha [41], which after adjusting for inflation (https://www.usinflationcalculator.com/) (last accessed 1 February 2022), equals USD 9392 in 2019. These total production costs include insecticide treatment costs. Based on average statistics on insecticide application costs in processing tomatoes in California with 22 different insecticides (https://www.cdfa.ca.gov/oefi/opca/docs/CDFA-neonic-report_2020_0729.pdf) (last accessed 1 February 2022), the average total costs (material and application costs) of insecticide treatments were equal to USD 113 per ha. Thus, USD 113 per ha is subtracted from USD 9392 per ha to obtain an estimate of non-insecticidal production costs (USD 9279). The profit potential is estimated by subtracting the non-insecticidal production costs (USD 9279) from the production value (USD 9813), which equals USD 534 per ha.

Regarding the damage rate by corn earworm on processing tomato, this pest is not present in California, but detailed data from experimental studies in Spain have been published [42]. Moreover, the authors generated linear regressions in 16 data sets of relationships between the percentage of damaged fruits by corn earworm larvae during crop growth and the percentage of damaged fruits at harvest. The different data sets represented infestations at different tomato phenological stages, two growing seasons, and infestations with different densities of corn earwom larvae. Although linear regressions were found to be statistically significant, the coefficient denoting slope varied considerably among the 16 data sets (0.23–0.73) [42]. However, this detailed study does support the argument of using a linear approach to model the damage rate. In the current scenario, it is assumed that the cropping universe equals 250 processing tomato plants, and it is assumed that each processing tomato plant can host four corn earworm larvae, so that carrying capacity, *K*, equals 1000. As damage rate, it is assumed that corn earworm densities of 0 and 1000 equal crop damages of 0% and 100%, respectively. Thus, crop damage rate per corn earworm larva equals 1.001. In some cases, the examined linear damage rates may be so high that, at high pest densities, the predicted cost of loss exceeds the economic value of yield potential. To address this, yield values (column L and starting in row 18 of the Excel spreadsheet) are calculated and presented based on a logical statement, in which 0 is the minimum yield.

Using the abovementioned values (Figure 4a), it is seen that there is a steep increase in resistance allele frequency after about 9–11 generations (Figure 4b) and that profit potential is high and constant for about nine seasons/generations but, afterwards, declines rapidly (Figure 4c). In this scenario, profit potential predictions in Figure 4c were based on an action threshold of EIL = 0, which implies that insecticides were applied in all generations (seasons). Based on analyses of nine insecticides, Torres-Vila, Rodríguez-Molina and Lacasa-Plasencia [42] recommended the use of an action threshold equal to 3% (applying insecticides when 3% of tomato plants showing signs of corn earworm damage). 

Figure 5a shows the predicted average profit for each of the first 15 seasons as a function of using different EIL values (0–6% of the carrying capacity, *K*). Figure 5b shows that an EIL of 2–4% of the carrying capacity was associated with a USD 3-higher profit per 250 tomato plants than pest management based on no threshold (EIL = 0%). Changing the damage rate, treatment costs, or profit potential and/or assuming different survival responses, fitness costs, and migration rates by genotypes would obviously lead to different model outputs. Additionally, it is very likely that a wide range of scenarios will suggest that an EIL = 0 is the most profitable option. Pest management based on a zero threshold may be especially advantageous if costs of insect pest monitoring (i.e., surveying traps or scouting crop plants) are high. Importantly, cost of monitoring is not parameterized separately in the interactive teaching tool, but it could be added to treatment costs. Additionally, a zero-threshold pest management strategy may be considered more advantageous, if the accuracy (risk of false positive (pest density falsely determined to be above EIL) and false negative (pest density falsely determined to be below EIL)) of pest density predictions is low and the cost/time consumption is taken into account.

## 4. Discussion

The effective teaching of complex and somewhat abstract concepts, such as insect population dynamics and insecticide resistance evolution, relies heavily on the ability to foster and facilitate an engaging teaching environments. Occasional articles discuss and present the effectiveness of teaching methods [43]. Some peer-reviewed journals, such as the *Journal of Visualised Experiments*, focus on the presentation of studies, but only a few studies published in these journals focus specifically on teaching tools. About a century ago, entomology instructors shared personal and almost anecdotal tales, which were clearly intended as ways to both reflect on their own shortcomings as teachers and on their hopes for improved teaching of entomology as a discipline in the future. It is in the same spirit that I share an interactive teaching tool with my peers. Furthermore, it is my hope that peer-reviewed entomology journals, such as *Insects*, will establish a new publication subject category for instructors to communicate and share freely available (open access) articles about teaching and about available teaching resources and ideas.

Population modeling is a cornerstone in applied entomology, and the first deterministic genetic population models of resistance evolution to insecticides were published 50 years ago [7,8,9], with the basic deterministic modelling framework remaining important today. In particular, many such modelling studies describe the evolutionary risk of resistance to genetically modified crops expressing insecticidal toxins, such as *Bt* (*Bacillus thuringiensis*) [13,35,36,38,44,45,46,47,48,49,50,51]. In addition to the large body of deterministic genetic population models of resistance evolution to Bt, there is a considerable number of studies describing resistance evolution to non-*Bt* insecticides [14,44,52,53,54,55,56,57]. Population modeling is a topic and skill set that requires both basic mathematical and biological knowledge, and it can only be taught effectively if both students and instructors have access to available programming platforms. The proposed interactive teaching tool could have been developed in R, C++, Python, Matlab, or other similar programming languages, but such versions would have required more advanced programming skills, software configurations (download and installation of specific libraries, etc.), and/or purchase of software licenses. Thus, although an Excel version may not be the most elegant solution (and does not readily enable features, such as running thousands of randomized simulations after adding stochasticity to coefficients), it was preferred due to the widespread availability and access to Excel via Microsoft Office.

Applications of insecticides and other pesticides in agricultural systems greatly alter pest genotype frequencies [58], and pesticide applications constitute the main approach used in pest management across agricultural systems [59]. Moreover, it has been estimated that, on an annual basis, at least 20 countries apply over 2 kg of insecticides per hectare of harvested agricultural land [59]. Agricultural systems are therefore highly suitable as models used to study evolutionary processes and population dynamics in response to management practices, such as pesticide applications [60]. The proposed genetic population model of resistance evolution and basic economics of pest management relies on a large number of important simplifications, so it may only have limited ability in predicting the outcomes of real-world (commercial) scenarios. However, as a teaching tool and to introduce students to a well-known and widely used model structure, the interactive teaching tool is believed to have considerable utility.

Through hands-on experience with this and similar interactive teaching tools, students acquire fundamental knowledge about the basic structures of population models, how models can be used to describe and visualize complex interactions and to produce outcomes, and students acquire experience with quantitative data interpretation. Furthermore, data from published studies can be analyzed and discussed using this interactive teaching tool.

## Figures and Tables

**Figure 1 insects-13-00169-f001:**
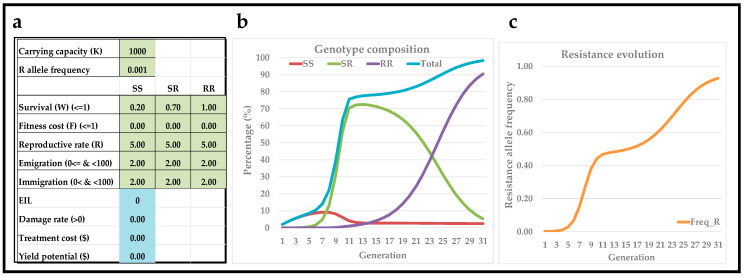
User interface (**a**) of the interactive teaching tool with input values in light-green (genetics) and light-blue (basic economics of pest management). Depictions of genotype composition (**b**) and resistance allele frequency (**c**) over 30 discrete and univoltine generations.

**Figure 2 insects-13-00169-f002:**
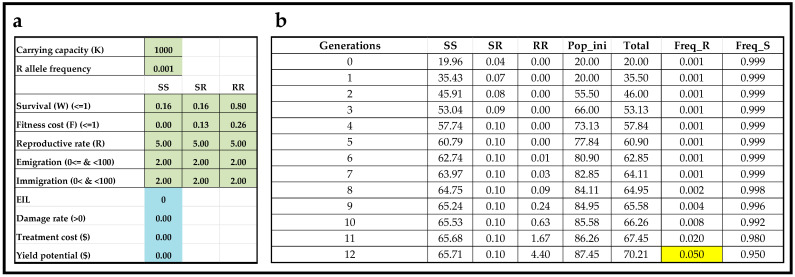
Resistance allele frequency after 12 generations with low immigration. User interface (**a**) of the interactive teaching tool with input values in light-green (genetics) and light-blue (basic economics of pest management). Table of resistance allele frequency reaching 5% after 12 discrete and univoltine generations (highlighted in yellow) (**b**).

**Figure 3 insects-13-00169-f003:**
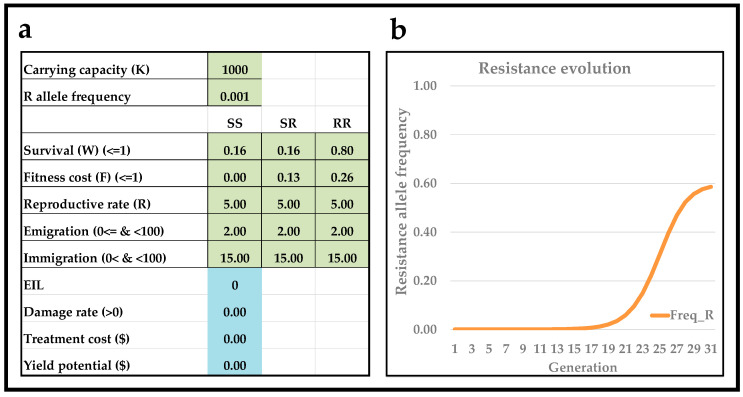
Resistance allele frequency with high immigration (presence of 15% non-*Bt* refuges). User interface (**a**) of the interactive teaching tool with input values in light-green (genetics) and light-blue (basic economics of pest management). Depiction of genotype composition over 30 discrete and univoltine generations (**b**).

**Figure 4 insects-13-00169-f004:**
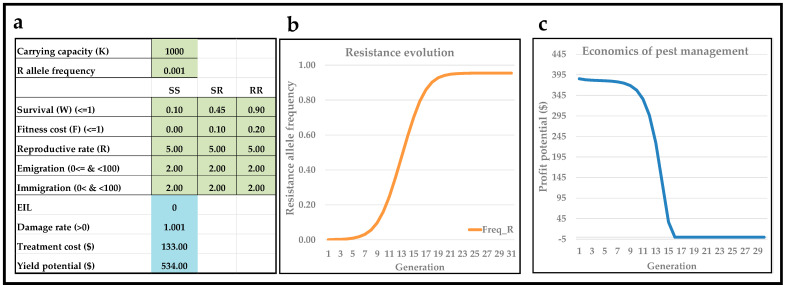
Profit potential with/without use of EIL (economic injury level) and with incomplete resistance. User interface (**a**) of the interactive teaching tool with input values in light-green (genetics) and light-blue (basic economics of pest management). Depictions of genotype composition (**b**) and profit potential (**c**) over 30 discrete and univoltine generations.

**Figure 5 insects-13-00169-f005:**
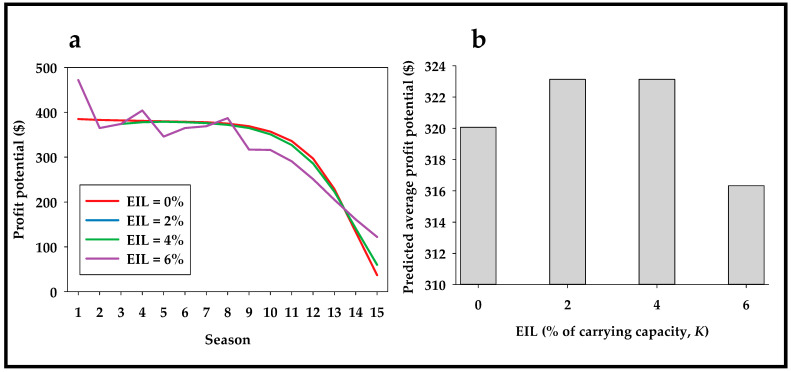
Predicted profit potential ($) as a function of EIL (economic injury level). Time series (**a**) and average (**b**) of profit potential for the first 15 discrete generations based on coefficients presented in Figure 4a but with different EIL (economic injury levels) ranging from 0–6% of carrying capacity (0–60 insect pest individuals as carrying capacity, K, equals 1000).

## Data Availability

This manuscript was produced without the inclusion of any actual data.

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
