# Peer review of "An Interactive Teaching Tool Describing Resistance Evolution and Basic Economics of Insecticide-Based Pest Management"

_insects, 2022, doi:10.3390/insects13020169_

Round 1

Reviewer 1 Report

In this work, a useful teaching tool for population genetics is presented. My recommendation is the acceptance after minor revision.

Some specific questions and comments

Simple Summary.

Line 6: This phrase is repeated three times (here, introduction and discussion). Please modify as it is a bit selective, the program seems to be aimed at a select group of students and not at everyone.

Lines 13-14: Both students and instructors are strongly encouraged to… Do not include in abstract but in material and methods or as a note at the end of the text.

Abstract

Line 40: merit? Perhaps more appropriately interest or utility.

Introduction

Line 57:  About the target…

Material and Methods.

Although it is later clarified in lines 106-107, before the equation (line 89), it would indicate that it is a bialelic locus.

Here you are talking about a locus with more than two allelic alternatives, what is the meaning of polygenic here? (line 109)

W is the letter denoting fitness, in this case survival is one more component of fitness along with the parameters R and F. In order not to confuse the students, I would designate the survival component with another letter.  Is it also a relative parameter?

Lines 119-121: resistance evolution after… I think it would be more correct to call it resistance evolution rate.

Immigration:

Both to clarify it in material and methods and in the results represented in Figure 1. What is the R allele frequency in the population of origin?

In the results there is a concrete reference to the absence of resistance to Bt corn in individuals coming from a reservoir.

Results:

Lines 249 and 251: There is an increase in resistance; increase instead of evolution.

Discussion

Lines 353-356: In my opinion, it is a value judgment, subjective and it has no place in this article or in this journal.

Author Response

Reviewer 1

Simple Summary.

Line 6: This phrase is repeated three times (here, introduction and discussion). Please modify as it is a bit selective, the program seems to be aimed at a select group of students and not at everyone. THE DISCUSSION HAS BEEN REVISED TO ADDRESS THIS ISSUE OF REDUNDANCY.

Lines 13-14: Both students and instructors are strongly encouraged to… Do not include in abstract but in material and methods or as a note at the end of the text. THAT IS A VERY GOOD SUGGESTION AND REVISIONS HAVE BEEN MDE ACCORDINGLY.

Abstract

Line 40: merit? Perhaps more appropriately interest or utility. THIS WAS REVISED.

Introduction

Line 57:  About the target… IT WAS UNCLEAR, WHAT THE REVIEWER IS SUGGESTING REVISION OF.

Material and Methods.

Although it is later clarified in lines 106-107, before the equation (line 89), it would indicate that it is a bialelic locus. THIS HAS BEEN INCLUDED

Here you are talking about a locus with more than two allelic alternatives, what is the meaning of polygenic here? (line 109). THIS VALID POINT HAS BEEN CLARIFIED.

W is the letter denoting fitness, in this case survival is one more component of fitness along with the parameters R and F. In order not to confuse the students, I would designate the survival component with another letter.  Is it also a relative parameter? LETTERS USED FOR EACH PARAMETER WERE BASED ON ONE OF THE ORIGINAL ARTICLES: Georghiou, G.P.; Taylor, C.E. Operational influences in the evolution of insecticide resistance. Journal of Economic Entomology 1977, 70, 653-658, doi:10.1093/jee/70.5.653.

Lines 119-121: resistance evolution after… I think it would be more correct to call it resistance evolution rate. THIS HAS BEEN CORRECTED.

Immigration: Both to clarify it in material and methods and in the results represented in Figure 1. What is the R allele frequency in the population of origin? EXCELLENT POINT – MEANING, ADDITIONAL EXPLANATION WAS DIRELY NEEDED AND HAS BEEN INCLUDED.

In the results there is a concrete reference to the absence of resistance to Bt corn in individuals coming from a reservoir. IT IS ASSUMED THAT THE VAST MAJORITY OF IMMIGRATING INSECT PEST INDIVIDUALS ARE “SS” – JUST AS IT IS ASSUMED UNDER THE HIGH-DOSAGE REFUGE STRATEGY. AS THE GENOTYPIC COMPOSITION OF IMMIGRATING INDIVIDUALS HAS NOW BEEN REVISED, I BELIEVE THE LANGUAGE IS MUCH CLEARER.

Results:

Lines 249 and 251: There is an increase in resistance; increase instead of evolution. THE TERM “RESISTANCE EVOLUTION” IS USED BECAUSE IT IMPLIES DEVELOPMENT OF RESISTANCE OVER TIME/GENERATIONS AND IT IMPLIES THAT RISK RESISTANCE OVER TIME IS UNDERPINNED BY THE SAME FUNDAMENTAL MECHANISMS AS OTHER GENETIC SELECTION PRESSURES.

Discussion

Lines 353-356: In my opinion, it is a value judgment, subjective and it has no place in this article or in this journal. UNFORTUNATELY, IT WA SUNCLEAR TO ME PRECISELY WHAT STATEMENT THE REVIEWER IS CONCERNED ABOUT?

Reviewer 2 Report

Dear author, 

Please find attached observations I have made concerning your manuscript.  The only major issue I have is that the link to the interactive tool/tutorials is broken. Worth checking it out to make sure it could be accessed easily. 

Author Response

Reviewer 2

Simple Summary:

Lines 12-13: I have tapped onto this link and copy-pasted it to get to this website. But the link is broken. Could author update this link or confirm if its my system not working? VERY UNFORTUNATELY IN THE ABSTRACT, THE JOURNAL TEMPLATE AUTOMATICALLY DIVIDED THE HYPERLINK (DUE TO LENGTH) AND PLACED A SPACEBAR, “-“, WHICH MESSED UP THE LINK. THE ACTUAL LINKS WORKS: https://chrnansen.wixsite.com/nansen2/teachingtool

Line 16: students will acquire and instructors who will read and benefit from this paper. THE SENTENCE WAS REVISED

Lines 15-19 Long: I AGREE AND HAVE REVISED ACCORDINGLY.

Abstract:

Introduction:

Line 123: on fitness cost, it is important for the author to provide more references. At least one each from agricultural pest and from a disease vector. SEVERAL REFERENCES HAVE BEEN ADDED.

Line 141: The fixed I-value, T to be capitalized. CORRECTED

Results:

Lines 201-202: I feel it is better to start with text introducing the results before author paste a figure. TO ACCOMMODATE THIS COMMENT, FIGURE 1 HAS BEEN MOVED DOWNWARDS, AND THE LAYOUT TEAM IS WELCOME TO PLACE IT WHEREVER IT IS CONSIDERED MOST SUITABLE.

Figure 1b and c lacks axes titles. Important to provide y and x axes for the excel. THIS IS AN EXCELLENT POINT, AND AXIS LEGENDS HAVE BEEN ADDED.

Line 212: Could one use this model with empirical data? For example, data of fitness cost associated with insecticide exposure? How could one prepare and input such a data. It would be helpful to describe with example a real life scenario of some inputs into the model. I BELIEVE SECTIONS 3.2. AND 3.3. ADDRESS THAT POINT AS I DRAW FROM PUBLISHED DATA TO GENERATE SCENARIOS OF BOTH PESTICIDE EVOLUTION AND OF ECONOMICS OF PEST MANAGEMENT PRACTICES.

Line 347 recommended: IT WAS UNCLEAR, WHAT THE REVIEWER IS SUGGESTING REVISION OF.

Discussion:

Lines 348-353: sounds repetitive as same from the introduction/simple summary/abstract. Please revise this to make different. THE REVIEWER RAISED A VALID POINT, AND I DECIDED TO MARKEDLY REVISE THE DISCUSSION.

Line 357: There are other platforms, which I am sure author know as well. For example, the journal of visualised experiment. But sometimes access is limited and one can only download the pdf if lucky. Author can discuss the problem of these type of platforms, further strengthening the need to have a dedicated, open access platforms for interactive teaching resources. THIS IS ANOTHER VALID AND INTERESTING POINT, AND THE DICUSSION WAS REVISED ACCORDINGLY.

Line 366: For R there is no need for any license. I will suggest to change this to reflect the need of having expertise of complex software to perform analysis. Most especially, authors can cite regions (like in Africa) where such knowledge of software like R and Python is limited. Important to separate R and python from Matlab. tutorials as well as inquiries about the interactive tool are welcome from both students and. THIS COMMENT IS VERY VALID AND ADDITIONAL LANGUAGE HAS BEEN ADDED

Line 379-383: repetition of previous sections. Should please be revised. THIS HAS BEEN REVISED.